# *BRAF* V600E Detection in Liquid Biopsies from Pediatric Central Nervous System Tumors

**DOI:** 10.3390/cancers12010066

**Published:** 2019-12-25

**Authors:** Noemi García-Romero, Josefa Carrión-Navarro, Pilar Areal-Hidalgo, Ana Ortiz de Mendivil, Adriá Asensi-Puig, Rodrigo Madurga, Rocio Núñez-Torres, Anna González-Neira, Cristobal Belda-Iniesta, Victor González-Rumayor, Blanca López-Ibor, Angel Ayuso-Sacido

**Affiliations:** 1Fundación de Investigación HM Hospitales, HM Hospitales, 28015 Madrid, Spain; noemigromero@gmail.com (N.G.-R.); peps86@gmail.com (J.C.-N.); pilarareal@hotmail.com (P.A.-H.); aomendivil@yahoo.es (A.O.d.M.); rmadurga@fundacionhm.com (R.M.); cbelda.hulp@salud.madrid.org (C.B.-I.); 2Pediatric Hematology and Oncology Unit, Madrid Montepríncipe Hospital, 28660 Madrid, Spain; 3Atrys Health, 08025 Barcelona, Spain; aasensi@atryshealth.com (A.A.-P.); vgrumayor@atryshealth.com (V.G.-R.); 4Spanish National Cancer Research Center (CNIO), 28029 Madrid, Spain; rnunez@cnio.es (R.N.-T.); agonzalez@cnio.es (A.G.-N.); 5Facultad de Medicina (IMMA), Universidad San Pablo-CEU, 28668 Madrid, Spain

**Keywords:** pediatric brain tumors, liquid biopsy, *BRAF*, dabrafenib

## Abstract

Pediatric Central Nervous System (CNS) tumors are the most fatal cancer diseases in childhood. Due to their localization and infiltrative nature, some tumor resections or biopsies are not feasible. In those cases, the use of minimally invasive methods as diagnostic, molecular marker detection, prognostic or monitoring therapies are emerging. The analysis of liquid biopsies which contain genetic information from the tumor has been much more widely explored in adults than in children. We compare the detection of *BRAF* V600E targetable mutation by digital-PCR from cell-free-DNA and EV-derived DNA (ctDNA) in serum, plasma and cerebrospinal fluid (CSF) isolated from a cohort of 29 CNS pediatric patients. Here we demonstrate that ctDNA isolated from serum and plasma could be successfully analyzed to obtain tumor genetic information which could be used to guide critical treatment decisions.

## 1. Introduction

Central Nervous System (CNS) tumors are the most common solid tumors and the most deadly forms of cancer in childhood [1], with an incidence of 5.64 cases per 100,000 people in the U.S. [2]. The 5-year survival rate for children depends not only on the tumor grade, but also on the location, being Diffuse Intrinsic Pontine Glioma (DIPG) the most aggressive, with an overall survival of less than 1 year [3]. Treatment of CNS tumors is based on a multimodal therapy including surgery, radiotherapy, chemotherapy and emergent adjuvants such as targeted therapy or immunotherapy [4]. The first step in brain tumor management is, if possible, to obtain a tumor sample in order to achieve the most accurate pathological and molecular diagnosis, being the extension of the surgical resection one of the most decisive factors in survival in many of the cases. However, some tumors, such as midline tumors, due to their anatomic location and infiltrative nature cannot undergo surgery, and biopsy is extremely difficult and risky [5]. In those cases the diagnosis is based only on clinical symptoms and Magnetic Resonance Imaging (MRI) [6]. For those tumors, the use of several strategies for biomarker discovery are emerging, such as the stereotactic-guided surgical biopsy [7], or the isolation of genetic material from liquid biopsies [8]. The last one is a non-invasive technique consisting of the isolation and study of material released from cancer cells to the circulating body fluids [9]. Although peripheral blood is the most frequent source of liquid biopsy, in the last few years the analysis of cerebrospinal fluid (CSF) from minimally invasive lumbar punctures is increasing [10,11]. These biofluids mainly contain circulating tumor cells (CTCs) and circulating DNA (ctDNA) which is composed of cell-free DNA (cfDNA) and extracellular vesicle (EV)-derived DNA [12]. The combined study of genetic and proteomic information obtained from this material shows a great potential for use in clinical practice [13].

The main advantage of liquid biopsy application in cancer disease is that it better reflects the biology of the entire tumor and the intra-tumor heterogeneity, guiding for an appropriate therapy or disease monitoring along tumor progression [14]. In this context, the development of techniques to detect targetable mutations offers enormous opportunities for childhood cancer patients. One of these alterations is the *BRAF* gene point mutation in the exon 15 at codon 600 resulting in an aminoacid replacement from valine to glutamic acid (BRAF p.V600E) [15], which causes a constitutive activation of the MAPK/ERK pathway [16]. This mutation appears in 33% of midline tumors, and has the highest incidence (66%) in pleomorphic xanthoastrocytomas [17]. The relevance of the *BRAF* V600E mutation presence in pediatric Low Grade Gliomas is that the 10-year progression-free survival rate decreases in almost 40% [18], however, the use of BRAF inhibitors as dabrafenib (GSK2118436), encorafenib (LGX818) and vemurafenib (PLX4032) have shown a successful response [19,20,21]. Although, several clinical trials in pediatric tumors are ongoing, dabrafenib is the most popular one used in clinical practice, as it has the highest effectivity in vitro and the highest brain distribution [22].

One of the most relevant challenges in clinical application of liquid biopsies is the development of methods that could detect low-allele-fraction variants [23]. Thus, digital PCR (dPCR) has been presented as an ultrasensitive technology capable of detecting and quantifying the rare allele with a high efficiency as compared with traditional methods [24].

In the present study we have isolated ctDNA, including EV-derived DNA, from a cohort of 29 pediatric cancer patients diagnosed with CNS tumors from serum, plasma and CSF. We show that ctDNA found in serum and plasma could provide the genetic characteristics of the original tumor. More interestingly from a therapeutic point of view, we have identified the actionable driver mutation *BRAF* V600E in both liquid biopsy sources. Its clinical application could be used to guide critical treatment decisions in those pediatric patients and other pathologies in which that mutation is quite predominant such as melanoma, colorectal carcinoma or papillary thyroid carcinoma [25].

## 2. Results

### 2.1. Isolation of ctDNA in Children Diagnosed with CNS Tumors

To explore the possibility of using ctDNA obtained from liquid biopsies as a biomarker source, we isolated serum, plasma and CSF from 29 pediatric patients. As lumbar puncture is not a risk-free invasive procedure, our CSF samples were only obtained from those patients in which its isolation was required for another medical procedure, such as the study of tumor cell dissemination or to explore CNS infections. Moreover, we sequenced DNA obtained from leukocyte suspension to discard germline mutations (Table 1 and Appendix A).

### 2.2. Evaluation of Mutation Status in Solid Tissue

At the time of diagnosis, solid tissue was used to sequence the *BRAF* gene and study its V600E mutation. From our cohort, 31.03% of the patients underwent biopsy (while total resection was only feasible in four of the patients due to their tumor’s localization. Four of 13 patients analyzed harbored the *BRAF* V600E mutation (Table 2).

### 2.3. Evaluation of BRAF V600E Mutation from Different Sources of Liquid Biopsies

We first determined the number of false positives of the assay using negative controls. The average of false positives was 3.25 mutant calls per 16,878 wild type calls. Therefore, we calculated the false positive ratio (RFP) and used it to calculate the expected false positive calls (ΛFP) for each sample. Finally, we calculated each p value as proposed by Milbury et al. [26] and used a 95% confidence threshold to reject the null hypothesis that the sample is wild type:*p* value = 1 − POISSON.DIST (NMut − 1, ΛFP, TRUE)

Then, we evaluated the ability to detect the *BRAF* sequence in ctDNA from the serum of all patients involved in the study, except for LI.18, which was not available (N = 28). We detected the *BRAF* V600E mutation in three patients (LI.31, LI.33 and LI.38). Then, we studied all plasma samples (N = 29) and identified the mutation in three thalamic astrocytoma patients (LI.10, LI.11 and LI.36). Unfortunately, no amplification was detected in 38.46% of ctDNA isolated from CSF, in which only two samples (LI.4 and LI.37) showed the presence of the mutation (Table 3). Surprisingly, for all the patients which gave a positive result, it could only be found in one of the liquid biopsy sources.

### 2.4. ctDNA Isolated from Blood-Liquid Biopsies Represents the Mutation Status from the Original Tumor

To further confirm the utility of liquid biopsies for biomarker detection, we analyzed the concordance between *BRAF* mutation status in the original tumor and in liquid biopsy. As only five paired samples were available between CSF and tissue, the study was performed in ctDNA isolated from serum and plasma. Based on dPCR results, ctDNA from serum represents better the *BRAF* mutation status in paired solid tumors rather than plasma. Genetic information found in plasma showed 25% sensitivity and 77.8% specificity, whereas ctDNA from serum had 50% sensitivity and 100% specificity for the detection of *BRAF* V600E mutation. Therefore, serum was found to be more accurately representative of the original tumor. However, our study findings must be interpreted with caution as a small cohort was analyzed. Notably, there were discrepancies in the samples from patient LI.25 whose solid biopsy was previously determined as mutant while dPCR results presented wild-type phenotype in serum and plasma samples (Table 4).

### 2.5. Response to the BRAF Inhibitor Dabrafenib in a Pediatric Patient with a BRAF V600E Mutated Cerebellar Ganglioglioma

To assess the clinical relevance in the detection of *BRAF* V600E mutation, we present a case report of the 2-year-old patient LI.31. This patient was diagnosed with unresectable cerebellar ganglioglioma in 2012 and progression of his disease was objectified despite treatment with chemotherapy and radiotherapy. Two years later, MRI images showed a large cerebellar hemisphere and brainstem tumor crossing midline with high signal intensity on the T2-weighted scan. Substantial mass effect was noted, with obliteration of fourth ventricle. The analysis of the solid biopsy revealed *BRAF* V600E mutation by conventional PCR. Thereby, patient started treatment with 2 mg/kg of dabrafenib twice daily. Follow-up imaging within 6 months showed a reduction in mass effect and tumor volume with a notable decrease in tumor enhancement on contrast-enhanced MRI images, consistent with a partial response based on RANO criteria. Nowadays, patient upholds five years of stable disease with maintenance therapy (Figure 1). As shown in Table 3, when liquid biopsies of this patient were analysed, only the serum sample demonstrated concordance with the results obtained in the previous solid biopsy, positive for *BRAF* V600E mutation, whereas plasma analysis revealed a WT mutational status.

## 3. Discussion

In the present study, we isolated cfDNA and EV-derived DNA from different types of liquid biopsies. Serum, plasma and CSF were obtained simultaneously from a cohort of pediatric patients (N = 29) diagnosed with CNS tumors, avoiding bias produced by the collected time [27]. We evaluated and compared the efficacy of those liquid biopsies as a source of diagnostic biomarkers for childhood brain tumors. Conventional PCR and subsequent sequencing revealed the presence of the *BRAF* V600E mutation in four solid biopsies at the time of diagnosis. These results corroborate previous studies showing that this mutation appears in the majority of pleomorphic xhantoastrocytomas (PXA) [17], as patient LI.25, and in 20% of gangliogliomas and pilocytic astrocytomas [28], which correspond with patients LI.31 and LI.38, respectively. The other mutated sample in our cohort is a pilomyxoid astrocytoma (LI.36) which is one of the most common gliomas in children [29]. In all cases, except for the PXA, *BRAF* V600E was confirmed by dPCR in serum or plasma. Although recent studies show that CSF is more representative of the original tumor genetic features than plasma in Glioblastoma patients [10], we cannot evaluate it, as we only have five paired samples whose solid biopsies have been analyzed for *BRAF* mutation. Other groups reported this similar limitation with ctDNA amplification problems in 33% of CSF samples depending on the CSF obtention procedure [30].

To the best of our knowledge this is the first study to compare different sources of liquid biopsies in pediatric cancers [31], an unmet need for clinical practice. Our data reveal that capturing ctDNA from blood-based biofluids such as serum and plasma could be used for biomarker detection by dPCR in pediatric patients. The main limitation of the test is the probability of finding false negatives within the samples, although they can also be found in the gold standard analysis as solid biopsies do not necessarily reflect the entire tumor clonal variability. Moreover, these results are only a proof of concept for the use of serum and plasma as biomarker sources in pediatric patients as additional studies with larger cohorts are needed to consolidate it.

Our *BRAF* V600E sensitivity values are slightly lower than those observed by other authors in other pathologies [32]. One possible explanation is that those tumors rarely metastasize outside the CNS [33], so ctDNA released to the bloodstream is restricted and makes minority allele detection more difficult [34]. Nevertheless, it could be useful when a solid biopsy is not available and for therapy response prediction [35], as has been approved by the FDA in NSCLC patients [9], in which values obtained for EGFR mutations were 100% of specificity and 65.7% of sensitivity [36].

*BRAF* V600E mutation analysis in cfDNA and EV-derived DNA found in serum could identify a subset of patients that could respond to certain target inhibitors, as we observed in the patient LI.31, improving the overall survival and progression-free survival. Moreover, it could help identify resistance mechanisms during therapy or monitor disease progression or relapse. In those patients, resistance occurs with the reestablishment of MAPK pathway or by the activation of PI3K/Akt signaling [37]. 

In some CNS tumors first biopsies or follow up re-biopsies are often unfeasible, so the analysis of liquid biopsies could overcome these difficulties and improve treatment personalization by screening for point mutations or measuring tumor burden [38].

## 4. Materials and Methods

### 4.1. Samples

Liquid biopsies including plasma, serum and CSF were obtained from 29 patients diagnosed with CNS tumors. CSF samples were collected by a lumbar puncture (LI.3, LI.4, LI.10, LI.18, LI.21, LI.26, LI.30, LI.37, LI.42, LI.43 and LI.44) or from an external ventricular device (LI.6, LI.11. and LI.22). Plasma and serum samples were obtained from peripheral blood samples by centrifugation. Buffy coats were isolated by Ficoll™ density gradient. Plasma, serum, buffy coat and CSF were stored at −80 °C. 

### 4.2. Circulating Tumor DNA Isolation and Quantification

500 µL of plasma, serum and CSF-derived circulating DNA was extracted with the QIAamp Circulating Nucleic Acid Kit (Qiagen, Hilden, Germany), as per the manufacturer’s protocol. Then, DNA concentration was determined using Qubit™ dsDNA High Sensitivity assay kit in a Qubit™ 3.0 fluorometer (ThermoFisher Scientific, Waltham, MA, USA) according to the manufacturer’s proceedings. DNeasy Blood and Tissue Kit (Qiagen) was used to obtain germline DNA.

### 4.3. Digital PCR

The digital PCR (dPCR) was performed according to the reference protocol for rare mutation using QuantStudio™ 3D Digital PCR System (ThermoFisher Scientific). We combined ctDNA with nuclease-free H_2_O, QuantStudio™ 3D Digital PCR Master Mix, and the ready-to-order TaqMan Assays (20X), ID: AH6R5PH for (*BRAF*) (ThermoFisher Scientific, Waltham, MA, USA)). Results were analyzed with QuantStudio™ 3D Analysis Suite Cloud Software (ThermoFisher Scientific) and the mean number of copies per µL was calculated.

### 4.4. Polymerase Chain Reaction (PCR)

For conventional PCR assays gDNA was amplified using these nucleotide sequences of the specific primers: Fw 5′CATGAAGACCTCACAGTAAA 3′ and Rv: 5′AGACAACTGTTCAAACTGAT 3′. PCR reactions were performed following the protocol of the Paq5000 enzyme (Agilent Technologies, Santa Clara, CA, USA). Thermal cycling conditions consisted of a 10 min denaturation step at 95 °C followed by 40 cycles at 95 °C for 20 s, 53.5 °C for 20 s and 72 °C for 30 s. Final PCR products were electrophoretically separated on 1.8% agarose gels.

### 4.5. Sequencing

Sequencing reactions were performed with BigDye^®^ Terminator v3.1 Cycle Sequencing Kit (ThermoFisher Scientific). Capillary electrophoresis was performed on a 3130 Genetic Analyzer (Applied Biosystems, Foster City, CA, USA). Sequencing reaction products were analysed with the Sequencing Analysis software (Applied Biosystems) and aligned using BLAST algorithm.

### 4.6. Magnetic Resonance Image (MRI) Data Acquisition and Analysis

Patient LI.31 underwent brain Magnetic Resonance Imaging at either 1.5 or 3 T on different magnets and using different protocols during the 5-year period of follow-up. Follow-up brain MRI for assessment of response was performed no less frequently than every 3 months. The MR imaging protocol used currently for brain tumor in our institution includes at least the sequences: axial T2 TSE/FSE 3 mm, axial Flair TSE/FSE 3–4 mm, diffusion weighted images (b = 0, 1000), 3D T1 MPRAGE/SPGR/FFE/TFE 1–1.5 mm with and without gadolinium contrast administration.

### 4.7. Study Approval

Samples from patients were kindly provided by the Pediatric Hematology and Oncology Unit-HM Hospitales (Madrid, Spain). Samples were processed following current procedures and frozen immediately after their reception. All patients participating in the study signed respective informed consent, and protocols were approved by institutional ethical committees (code CEIM HM-Hospitales: 18.12.1337-GHM).

### 4.8. Statistical Calculations

To assess the significance of the detection of *BRAF* V600E mutation by dPCR we obtained the average ratio of false positives from negative control samples, R_FP_ = 1.93 × 10^−4^. This ratio was used to obtain the average false positives, Λ_FP_, expected for each sample. Finally, a Poisson distribution with mean Λ_FP_ was used to obtain the significance of the observed positive calls for each sample. The null hypothesis that observed positive calls are false positives was rejected with a confidence threshold of 95%.

## 5. Conclusions

One of the main challenges in clinical practice is to ensure high accuracy for biomarker detection. Although further studies increasing the number of CSF derived ctDNA are required, we found serum as the most promising alternative rather than plasma for *BRAF* V600E by dPCR detection in liquid biopsy from CNS pediatric cancers. Therefore, a robust validation with samples from a large cohort is required to turn it into a validated clinical tool.

## Figures and Tables

**Figure 1 cancers-12-00066-f001:**
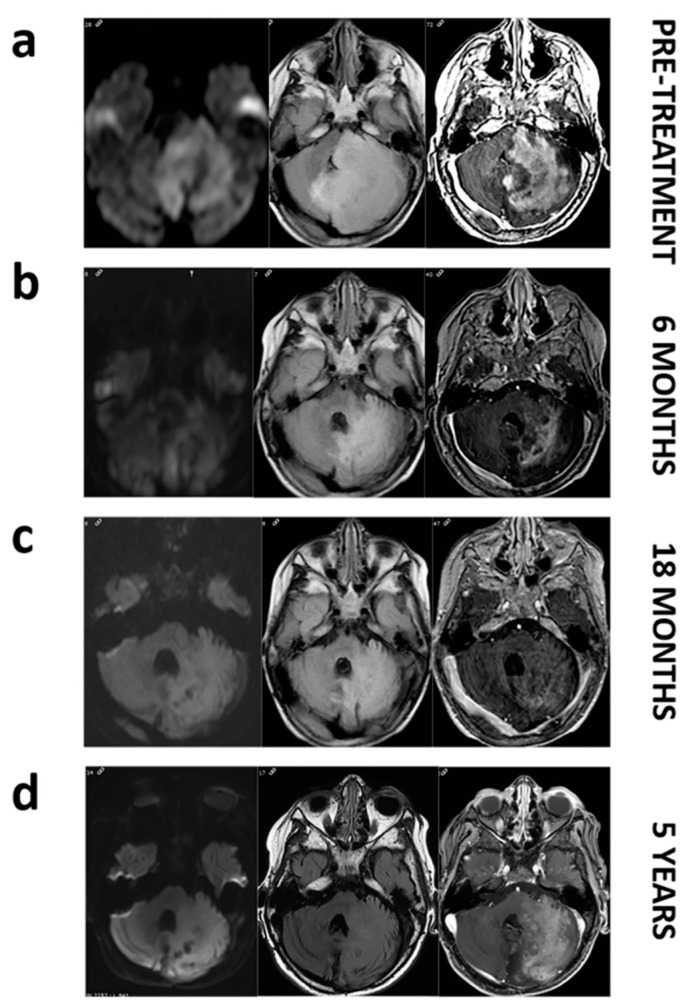
MRI images of patient LI.31 before and during dabrafenib treatment. Pretreatment images (**a**). Axial FLAIR image shows a large mass affecting left cerebellar hemisphere and pons with a marked mass effect on fourth ventricle. Axial diffusion (DWI) image reveals restricted diffusion within the mass. Axial T1 postcontrast image depicts intense enhancement. Follow up at 6, and 18 months (**b**,**c**). Axial FLAIR, DWI and post-contrast demonstrate a decrease in the mass extension and in the mass effect with a progressively enhanced shrinkage. Follow-up at 5 years (**d**). We do not notice a regrowth on axial FLAIR and even if enhancement reappears, restricted diffusion remains absent.

**Table 1 cancers-12-00066-t001:** Clinical features of the CNS patients and liquid biopsy source. *BRAF* V600E analysis by PCR in germline DNA.

Patient ID	Sex	Age	Pathological Diagnosis	Liquid Biopsy Source			Germline DNA
				Serum	Plasma	CSF	
LI.3	M	5	Medulloblastoma SHH Subtype	+	+	+	SN
LI.4	F	7	Anaplasic Medulloblastoma	+	+	+	WT
LI.6	F	17	Unresectable Thalamic Anaplastic Glioma	+	+	+	WT
LI.10	M	3	Thalamic Diffuse Astrocytoma	+	+	+	WT
LI.11	F	17	Unresectable Thalamic Pilocytic Astrocytoma	+	+	+	WT
LI.15	M	7	DIPG	+	+		WT
LI.17	M	6	Ependymoma	+	+		WT
LI.18	F	15	DIPG		+	+	WT
LI.21	M	6	Diffuse Glioma	+	+	+	WT
LI.22	F	2	Pilomyxoid Midline Astrocytoma	+	+	+	WT
LI.23	M	8	DIPG	+	+		WT
LI.25	M	8	Pleomorphic Xanthoastrocytoma	+	+		WT
LI.26	F	15	Oligodendroglioma	+	+	+	WT
LI.28	M	7	DIPG	+	+		WT
LI.29	-	9	DIPG	+	+		WT
LI.30	M	12	Thalamic Diffuse Glioma	+	+	+	WT
LI.31	M	2	Cerebellar Ganglioglioma	+	+		WT
LI.33	M	10	DIPG	+	+		WT
LI.34	F	4	DIPG	+	+		WT
LI.35	F	6	DIPG	+	+		WT
LI.36	M	6	Pilomyxoid Thalamic Astrocytoma	+	+		WT
LI.37	M	18	Classic Medulloblastoma	+	+	+	WT
LI.38	M	11	Pilocytic Astrocytoma	+	+		WT
LI.39	M	3	Supratentorial PNET	+	+		SN
LI.40	M	15	Pilocytic Astrocytoma	+	+		WT
LI.41	M	14	GBM	+	+		WT
LI.42	M	1	GBM	+	+	+	WT
LI.43	M	12	Diffuse Midline Glioma	+	+	+	WT
LI.44	F	21	Pilocytic Astrocytoma	+	+	+	WT

SN: Sample Not Available. WT: Wild Type.

**Table 2 cancers-12-00066-t002:** *BRAF* V600E Mutation analysis in solid biopsy (*N* = 29).

Sample	Number	%
Biopsy	9	31.03
Partial resection	5	17.24
Subtotal resection	5	17.24
Total resection	4	13.79
Missing	6	20.69
**BRAF**		
Mutant (V600E)	4	13.79
WT	9	31.03
Analysis not performed	16	55.17

**Table 3 cancers-12-00066-t003:** Analysis of *BRAF* V600E from ctDNA obtained from serum, plasma, CSF and original tumor.

	*BRAF* V600E				*p* Value		
Patient ID	Serum ctDNA	Plasma ctDNA	CSF ctDNA	Solid Biopsy	Serum ctDNA	Plasma ctDNA	CSF ctDNA
LI.3	WT	WT	WT	SN	5.75 × 10^−1^	3.89 × 10^−1^	5.08 × 10^−1^
LI.4	NA	WT	MUT	WT	NA	4.71 × 10^−1^	9.13 × 10^−3^
LI.6	WT	WT	NA	WT	1	9.50 × 10^−1^	NA
LI.10	WT	MUT	NA	WT	4.94 × 10^−1^	2.10 × 10^−1^	NA
LI.11	WT	MUT	WT	WT	7.69 × 10^−1^	2.56 × 10^−1^	6.66 × 10^−1^
LI.15	WT	WT	SN	WT	2.64 × 10^−1^	3.15 × 10^−1^	-
LI.17	WT	WT	SN	SN	5.91 × 10^−1^	1	-
LI.18	SN	WT	NA	SN	-	2.30 × 10^−1^	NA
LI.21	WT	WT	WT	SN	5.96 × 10^−1^	6.17 × 10^−1^	8.87 × 10^−1^
LI.22	WT	WT	WT	WT	7.58 × 10^−1^	9.34 × 10^−1^	9.46 × 10^−1^
LI.23	WT	WT	SN	SN	3.09 × 10^−1^	5.40 × 10^−1^	-
LI.25	WT	WT	SN	MUT	8.27 × 10^−1^	9.54 × 10^−1^	-
LI.26	WT	WT	NA	SN	6.20 × 10^−1^	7.23 × 10^−1^	NA
LI.28	WT	WT	SN	SN	5.78 × 10^−1^	9.51 × 10^−1^	-
LI.29	WT	WT	SN	SN	7.74 × 10^−1^	5.98 × 10^−1^	
LI.30	WT	WT	WT	SN	8.74 × 10^−1^	8.00 × 10^−1^	9.48 × 10^−1^
LI.31	MUT	WT	SN	MUT	1.74 × 10^−2^	4.50 × 10^−1^	-
LI.33	MUT	WT	SN	SN	4.64 × 10^−3^	2.88 × 10^−1^	-
LI.34	WT	WT	SN	SN	8.47 × 10^−1^	2.93 × 10^−1^	-
LI.35	WT	WT	SN	SN	3.60 × 10^−1^	1.17 × 10^−1^	-
LI.36	WT	MUT	SN	MUT	3.64 × 10^−1^	6.72 × 10^−4^	-
LI.37	WT	WT	MUT	SN	3.85 × 10^−1^	5.93 × 10^−1^	2.17 × 10^−8^
LI.38	MUT	WT	SN	MUT	2.33 × 10^−4^	5.02 × 10^−1^	-
LI.39	WT	WT	SN	SN	4.61 × 10^−1^	5.13 × 10^−1^	-
LI.40	WT	WT	SN	SN	8.76 × 10^−2^	4.92 × 10^−1^	-
LI.41	WT	WT	SN	WT	4.91 × 10^−1^	9.56 × 10^−1^	-
LI.42	WT	WT	WT	WT	6.38 × 10^−1^	5.24 × 10^−1^	6.85 × 10^−1^
LI.43	WT	WT	WT	WT	2.73 × 10^−1^	2.96 × 10^−1^	4.03 × 10^−1^
LI.44	WT	WT	NA	SN	8.81 × 10^−1^	7.23 × 10^−2^	NA

NA: Not Amplified; SN: Sample Not Available.

**Table 4 cancers-12-00066-t004:** Concordance between solid biopsy analysis and liquid biopsy (N = 13).

			Solid Biopsy						Accuracy
			Mutant	WT	Sensitivity	Specificity	PPV	NPV	
**Liquid Biopsy**	Plasma	Mutant	1	2	25%	77.8%	33.3%	70%	61.5%
		WT	3	7					
		Total	4	9					
	Serum	Mutant	2	0	50%	100%	100%	80%	83.3%
		WT	2	8					
		Total	4	8					

PPV: positive predictive value, NPV: negative predictive value.

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
