# Peer review of "BRAF V600E Detection in Liquid Biopsies from Pediatric Central Nervous System Tumors"

_cancers, 2019, doi:10.3390/cancers12010066_

Round 1

Reviewer 1 Report

Given the number of cases involved, I believe that the statement that serum is better than plasma is too strong in spite of the statistical analysis.  One case one way or the other would change the statistics.  I would soften the statement

Author Response

Following their advices, we have modified some sentences in the abstract, introduction, results and discussion sections, focusing in that both blood-liquid biopsies could provide genetic information from the original tumor.

Abstract section: “Here we demonstrate that ctDNA isolated from serum and plasma could be analyzed to obtain tumor genetic information, which could be used to guide critical treatment decisions “.

Introduction section: “We show that ctDNA found in serum and plasma could provide the genetic characteristics of the original tumor. More interestingly from therapeutic point of view, we have identified the actionable driver mutation BRAF V600E in both liquid biopsy sources”.

Results section:  We have modified the heading sentence 2.4.: “ctDNA isolated from blood-liquid biopsies represents the mutation status from the original tumor”.

And the following paragraph: “Therefore, serum was found to be more accurately representative of the original tumor. However, our study findings must be interpreted with caution as small cohort was analyzed”.

Discussion section: “Our data reveal that capturing ctDNA from blood-based biofluids such as serum and plasma could be used for biomarker detection by dPCR in pediatric patients”. “Moreover, these results are only a proof of concept for the use of serum and plasma as biomarker sources in pediatric patients as additional studies with larger cohorts are needed to consolidate it”.

Additioanly, we have reviewed the manuscript and corrected some typos and language related mistake.

We really thank Reviewer #1 and #2 for their observations that have significantly improved the quality of the manuscript.

Reviewer 2 Report

With such a small cohort, the authors make the conclusion that serum ctDNA is superior to plasma. First, this is not a justified claim given the incredible small n in table 4 (for example, sensitivity is based off of 4 samples). Second, the authors provide no explanation for why they report serum may be superior. In addition, there is no clear methodology describing what DNA is being analyzed in the serum.

To address this major critique, the authors should more simply state that this proof-of-principle study demonstrates that liquid biopsies with plasma and serum cell-free DNA can provide tumor-specific genetic data. There should be no assertion about superiority, though the authors can postulate why serum v plasma may be a more ideal analyte for future study.

Author Response

Following their advices, we have modified some sentences in the abstract, introduction, results and discussion sections, focusing in that both blood-liquid biopsies could provide genetic information from the original tumor.

Abstract section: “Here we demonstrate that ctDNA isolated from serum and plasma could be analyzed to obtain tumor genetic information, which could be used to guide critical treatment decisions “.

Introduction section: “We show that ctDNA found in serum and plasma could provide the genetic characteristics of the original tumor. More interestingly from therapeutic point of view, we have identified the actionable driver mutation BRAF V600E in both liquid biopsy sources”.

Results section:  We have modified the heading sentence 2.4.: “ctDNA isolated from blood-liquid biopsies represents the mutation status from the original tumor”.

And the following paragraph: “Therefore, serum was found to be more accurately representative of the original tumor. However, our study findings must be interpreted with caution as small cohort was analyzed”.

Discussion section: “Our data reveal that capturing ctDNA from blood-based biofluids such as serum and plasma could be used for biomarker detection by dPCR in pediatric patients”. “Moreover, these results are only a proof of concept for the use of serum and plasma as biomarker sources in pediatric patients as additional studies with larger cohorts are needed to consolidate it”.

We have also reviewed the English and corrected typos and mistakes.

We really thank Reviewer #1 and #2 for their observations that have significantly improved the quality of the manuscript.

This manuscript is a resubmission of an earlier submission. The following is a list of the peer review reports and author responses from that submission.

Round 1

Reviewer 1 Report

This is an interesting and informative study about the use of liquid biopsy techniques to detect BRAF mutations in pediatric CNS tumors. The authors note that there is not concordance between testing modalities, and though the number of patients is small, they do also identify generally consistent BRAF mutants across testing modalities when they had the ability to compare matched samples.

Together, this is a manuscript worthy of publication, though there are a few critiques that should be addressed:

The case described should be further evolved, particularly as it relates to liquid biopsy analysis. Without any connection to liquid biopsy techniques, and how this specifically was used in a clinically meaningful way, the case report does not contribute to this manuscript. Perhaps serial samples were collected and tracked for the mutant? Or, though not clearly stated, was the liquid biopsy positive for the BRAF V600E whereas the solid tumor profiling was not?

Fig 1. If the shaded right-angle triangle to the right is meant to reflect elapsed time, it is redundant and non-contributory. If it represents another metric, please describe.

The discussion lacks mention of a primary limitation: this study is small in size and is generally a proof of principle study.

Reviewer 2 Report

The authors state several times that they isolate cell-free DNA and EV-derived DNA but they actually never isolate EV from the liquid biopsies, so that statement is quite misleading. It would be interesting thought to know what the contribution of EV to the ctDNA content is in serum, plasma and CSF, and whether it differs.

The authors start wanting to compare serum, plasma and CSF but eventually the CSF data can not be used for any comparison because they do not have enough samples to match the patients carrying mutations. Expanding the cohort so that CSF could be included in the study will strengthen their results

Reviewer 3 Report

The authors present the results of analysis of circulating tumor DNA (ctDNA) in a series of 29 pediatric subjects with a variety of central nervous system (CNS) tumors.  The authors analyzed samples from plasma, serum, cerebrospinal fluid (CSF) and tumor. Specifically the authors screened for the presence of the BRAF V600E mutation which can be seen in up to one third of pediatric brain tumors.   The authors analyzed 16 samples from either biopsy or resection of tumor, 14 samples of CSF, 28 serum samples and 29 plasma samples.  They found the presence of the BRAF mutation in 2/14 CSF samples, 3/28 serum samples, 3/29 plasma samples and in 4 of 16 tumor samples.  There was no concordance between the positive serum, plasma or CSF samples.  There was concordance in 1/3 positive plasma samples with positive tumor sample.  The other two samples, the tumor had a wild type genotype (WT).Two of three serum positive samples were concordant with the findings in tumor and tumor was not available for the third sample comparison.  One positive sample from CSF was discordant with the tumor and tumor was not available for the comparison of the other case.  In six tumor samples with WT there was concordance with the samples obtained by serum, plasma or CSF.  On case of tumor mutant genotype was not detected by any of the "liquid biopsy" samples. The authors conclude that serum is a promising alternative for liquid biopsy rather than plasma.

Comments

I found it hard to make any statements regarding the result of the study given the number of samples and the number of positive samples. The only conclusion I could draw is that it is possible to detect the BRAF V600E mutation in liquid biopsies from children with CNS tumors. The liquid biopsies detected a total of 8 samples with mutations.  Some discordant with the tumor or when the tumor was not available.   Three had concordance of the finding. I find this an interesting result that the authors do not comment upon.  Is the combination of liquid biopsies from all three sources a better parameter to guide the clinician searching for targeted therapy? Given the low number of positive samples and low numbers overall I do not believe that you can make any statement regarding the superiority of any of the sources of liquid biopsy.  Rather I believe, as stated above, the combination of all three seems to yield more interesting results I would like to see the authors interpretation or opinions regarding those samples with positive liquid biopsy but WT tumors.